# Shaping a View on the Influence of Technologies on Sustainable Tourism

**Sandra Maria Correia Loureiro *** and **Jorge Nascimento**

Business Research Unit (BRU-IUL), ISCTE—Instituto Universitário de Lisboa, 1649-026 Lisboa, Portugal; Jorge_Nascimento@iscte-iul.pt
**\*** Correspondence: sandramloureiro@netcabo.pt

**Abstract:** To date, tourism is the fastest growing industry globally, but one of the least developed in terms of environmentally sustainable practices. However, only a small portion of documents elaborate on how the introduction of new technologies can impact a more sustainable development route for tourism. This study's objective is to provide an overview on literature state-of-the-art related to sustainable tourism and technological innovations, offering insights for further advancing this domain. We employ a bibliometric analysis and a comprehensive review of 139 articles, collected from Web of Science and Scopus databases, for the purpose of: (i) exploring and discussing the most relevant contributions in the publication network: (ii) highlighting key issues and emerging topics; (iii) uncovering open questions for the future. Our findings reveal contradictory views on the risks and benefits of technology adoption. Artificial intelligence, internet of things, circular economy, big data, augmented and virtual reality emerge as major trends. Five work streams are identified and described, leading to a broader perspective on how technology can shape the future of sustainable tourism. Relevant theoretical and managerial implications are derived. Finally, a research agenda is proposed as guidance for future studies addressing the outcomes of digital disruption on sustainable tourism.

**Keywords:** bibliometric analysis; eco-tourism; sustainability; tourism; artificial intelligence; virtual reality; augmented reality; research agenda; future trends



## 1. Introduction

When the COVID-19 pandemic erupted, Hospitality and Tourism (H&T) was the largest and fastest growing industry worldwide [1,2], accountable for substantial environmental impacts, related to water consumption, carbon footprint and waste generation, among others, and overall pressure on resources conservation. Nowadays, even though green lodging and eco-tourism are gaining preference among travelers [3,4], and are essential for the sector's future success, tourism is considered one of the least developed industries with regard to the implementation of sustainable practices [5,6]. The development of sustainable tourism is of utmost importance, especially in the post-pandemic period, considering the severe economic challenges it is facing, as well as the environmental crisis and climate changes experienced globally [7]. Sustainable tourism is defined by [8] as "*tourism that takes full account of its current and future economic, social and environmental impacts, addressing the needs of visitors, the industry, the environment and host communities*", pointing out toward the three dimensions of sustainability: economic, social, and environmental. For exploring this topic, concepts such 'green' and 'eco-tourism' will be explored to focus on the environmental aspects of sustainable development in tourism.

The most recent technological advances are already disrupting even the most traditional markets and can enable strategically agile processes. The effects on sustainability can be both positive and negative [4], affecting the achievement of United Nations' Sustainable Development Goals (SDG), and a more fairly distributed economic prosperity [9]. Conversely, while some authors examining the H&T sector highlight the lack of studies on how

the introduction of AI-enabled solutions, enriched virtual interactions and social robots will change the engagement with customers and employees in the future [10,11]; others argue how the fourth technological revolution is already altering the paradigm of the H&T industry [12,13], which includes breakthrough innovations, such as Virtual Reality (VR), Augmented Reality (AR), Artificial Intelligence (AI) [14,15], and the benefits of mobile interactivity for H&T customer engagement [16].

VR is an immersive 3D-simulated setting that allows consumers to have the feeling of being in a real-world environment [17]. The expression was originally coined by Jaron Lanier in the 1980s, leading to the invention of virtual reality gears, such as the Dataglove and the EyePhone head-mounted display [18,19]. AR relates purely virtual to purely real environments, where the observer is seeing the real world and can also visualize virtual objects overlaid on it, usually by wearing see-through displays, or interacting with their own mobile devices. VR and AR can both be used to promote a touristic destination or site, providing an immersive stimulation to tourists, for a totally new, memorable experience [18,20,21].

With regard to AI, the lack of a consensual definition has not prevented the spread of research about its new applications [22], where various definitions of AI systems are summarized into four categories along two dimensions: *reasoning–behaviour dimension* and *human performance–rationality dimension*. These are: (1) systems that think like humans, (2) systems that act as humans, (3) systems that think rationally, and (4) systems that act rationally. Authors elaborate on the exciting capabilities of AI systems, and report on four different levels of intelligence (e.g., *mechanical, analytical, intuitive, empathetic*), as AI-enabled entities evolve over time, with increased potential to transform society and organisations [23], exhibiting new skills, such as the ability to process and communicate in natural language, to store information and use it to answer questions, or *machine learning* (e.g., ability to adapt to new circumstances, detect and extrapolate patterns).

The present study intends to contribute towards consolidating a view of how digital innovations can influence the upcoming challenges of sustainable tourism. For that effect, a bibliometric analysis was conducted, followed by a comprehensive review of the latest, most pertinent literature, in order to achieve the following research goals: (i) explore the literature network and identify the main contributions, in the domains of environmentally sustainable tourism and AR/VR/AI technological innovations; (ii) understand and describe the intellectual structure of the current corpus of literature; (iii) propose a future research agenda towards the development of knowledge of how technologies can influence sustainable tourism. This investigation takes an unprecedented approach in this domain, by combining bibliometric analysis with a systematic review of the most recent and meaningful trends in literature, related to the opportunities and dilemmas of breakthrough technologies, its outcomes for consumers, workers, organisations, and nations, as well as the main issues for the sustainable development of tourism.

While doing so, we have found that these fields of research do not yet intercept each other, with a lack of understanding about how technologies can impact on the H&T industry, and how digital innovations can promote sustainable development. By exploring and combining non-related research streams, this paper opens a new set of research questions for the future, and advises on possible impediments, which can be used to foster collaboration across currently unrelated fields, energizing new routes for addressing the above-mentioned knowledge gap.

We will address the following research questions (RQs). RQ1: How is the collaborative network of authors, countries of affiliation, and documents in this field? RQ2: What are the most influential publications and the network composition of journals? RQ3: Which are the most popular topics, and their main context, among scholars? RQ4: What are the intellectual structure and major streams at the forefront of research on technology, tourism, and environmental sustainability? RQ5: What are the emerging trends and future research questions towards a better understanding of the influence of technology on the development of sustainable tourism?

This paper is structured as follows; in the next section, we shortly present the research method, followed by report and discussion of the findings from our analysis. Finally, we provide an overview of theoretical contributions and managerial implications for sustainable tourism, as well as the limitations of this study.

## 2. Materials and Methods

Bibliometric analysis was employed to identify and examine the main contributions in literature, exploring the adoption of new technologies and their impacts on sustainable tourism. Scholars in the field of bibliometrics apply quantitative techniques to the measurement of bibliographic data and scientific activity [24], which has emerged as a legitimate method, extensively used across a wide variety of disciplines [25]. Science mapping was used to extract the relationships between research constituents, that is, co-authorship, bibliographic coupling, co-word, citation analysis and co-citation analysis [26] and present the structure of knowledge on the interface of technology in sustainable tourism.

The evolution in bibliometric computation and visualization software allows researchers to explore large datasets efficiently. VOSviewer is an intuitive, open-source tool, which was selected for the present study. Widely recognized among scholars, VOSviewer was designed for constructing and viewing bibliometric maps, and used here to perform the bibliometric analysis [27].

### 2.1. Seach Protocol and Data Collection

Data were extracted on 11 September 2021. The articles were collected from two online databases, Web of Science (WOS) and Scopus, applying the following query to title, abstract and keywords: ((“virtual reality” OR “augmented reality” OR “artificial intelligence” OR “artificial-intelligence”) AND (“sustainab*” OR “eco-touris*” “eco touris*” OR “green touris*”)). The search strings were developed based on previous encompassing studies available in recent literature [7,18].

WOS displayed a set of 2205 documents, which was reduced to 174 documents when considering business, management, economics, and environmental studies as subject areas of appreciation, from which 138 are English-written articles. The same query in Scopus—applied to title, abstract and keywords—shows 13 documents, but only one of which is an article associated with the present research problem (Figure 1).

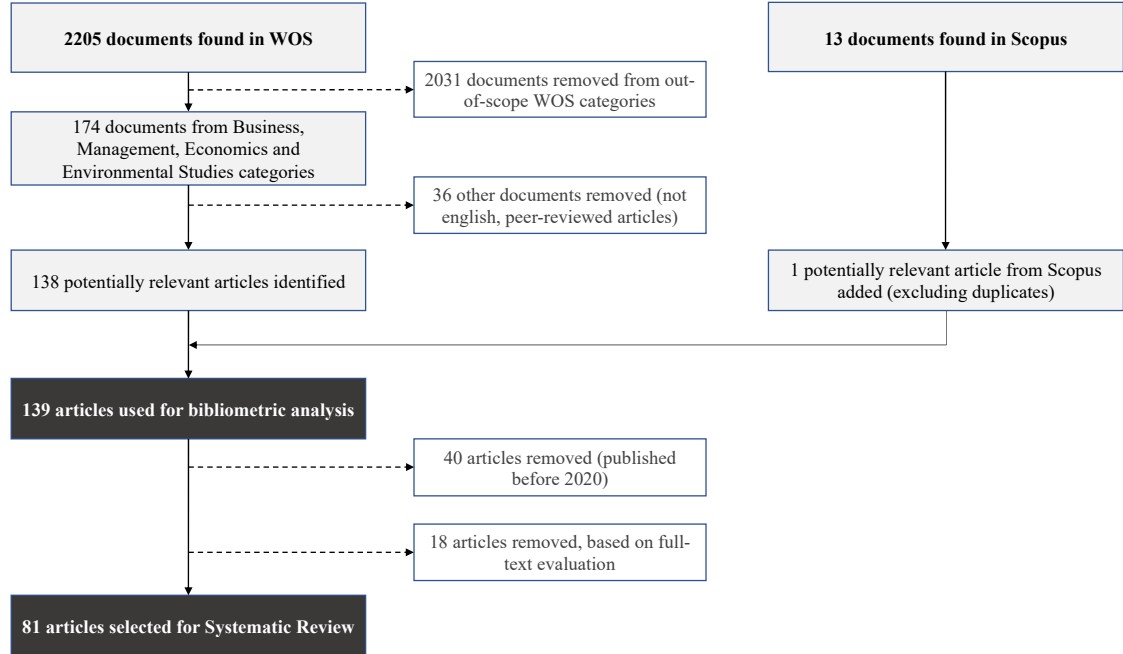

**Figure 1.** Search strategy and selection criteria.

Afterwards, further selection criteria were employed to the dataset, to select a core subset of articles, more related to our research problem and more recently published, which could represent the state-of-the-art of literature. For that matter, the number of documents was first reduced to 99—by retaining only publications from 2020 (inclusive) onwards—and then an additional 18 papers were excluded, based on: (a) not being directly related to the research theme (e.g., more technical aspects of AR/VR/AI development and implementation); (b) insufficient practical and theoretical contribution; (c) inconsistent theory-method-data link, or incomplete data; (d) weak development of literature. These criteria were adapted from literature [7].

### 2.2. Data Analysis

Citation, co-citation, co-authorship, bibliographic coupling, and co-word analysis were used to examine relationships among research constituents. Full counting method was applied. Co-authorship and bibliographic coupling were employed for examining the collaborative network (RQ1) of authors, respectively, and their respective countries of affiliation and publications. Citation analysis of the most popular articles, and co-citation analysis of the network composition of journals, were deployed to identify the most influential elements in the network (RQ2), followed by co-word analysis of keyword co-occurrence, to reveal the most popular topics of attention (RQ3) in the research community. The intellectual structure and major streams (RQ4), as well as emerging trends for the future (RQ5), derive from the systematic review of the selected articles.

### 3. Results

### 3.1. Cooperative Network of Authors, Countries of Affiliation and Documents

Co-authorship analysis illustrates the intellectual collaborations among academics (see Figure 2), where we can find three distinct cooperative networks. One group (red colour) comprises authors located in Australia [28,29] dealing with the use of immersive technologies (e.g., VR) for destination experiences, in situations of over-tourism leading to deterioration of the sites. Another group (in green), aggregates researchers from different countries, such as USA, France, or England, [30–33], and deals with AI for sustainable purposes at a firm level but considering different external stakeholders. The blue group also examines AI and other related technologies, for sustainable issues (e.g., France, England, Australia, Malaysia), but more focused in internal stakeholders [34,35].

Figures 3 and 4 display, respectively, the network of universities' countries of origin, from where the first authors are affiliated to, and the network of documents. When applying bibliometric coupling technique to countries of affiliation, four major networks emerge: the two in red and yellow colours link European countries, while the other two extend their connections across universities from different continents. This analysis is conducted based on the assumption of similarity between two articles that sharing common references [26]. Regarding the documents' bibliometric coupling (articles published in indexed journals), Figure 4 highlights the articles' proximity—due to sharing similar content—and recency, where these examples exhibit a central, closely related position [36–38].

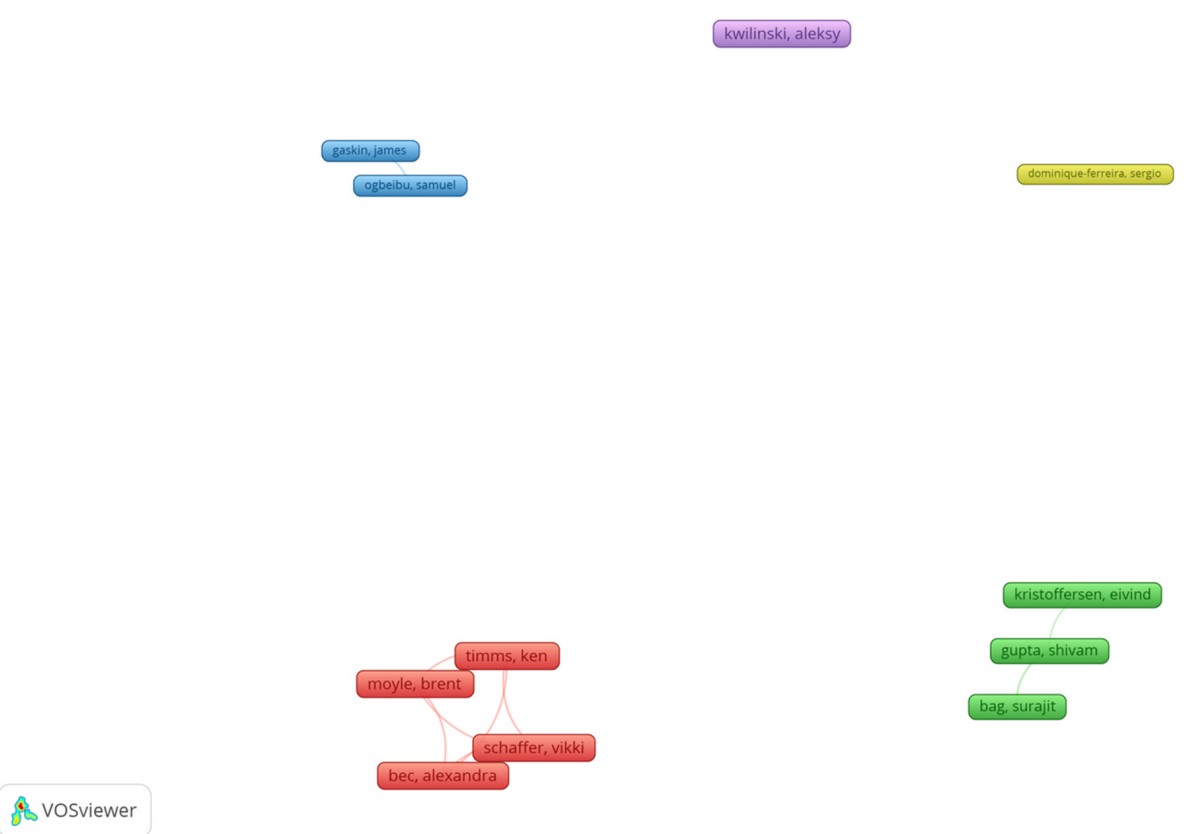

**Figure 2.** Cooperative network of authors. Note: The colours show the main networks established. The size of the frame represents the number of citations.

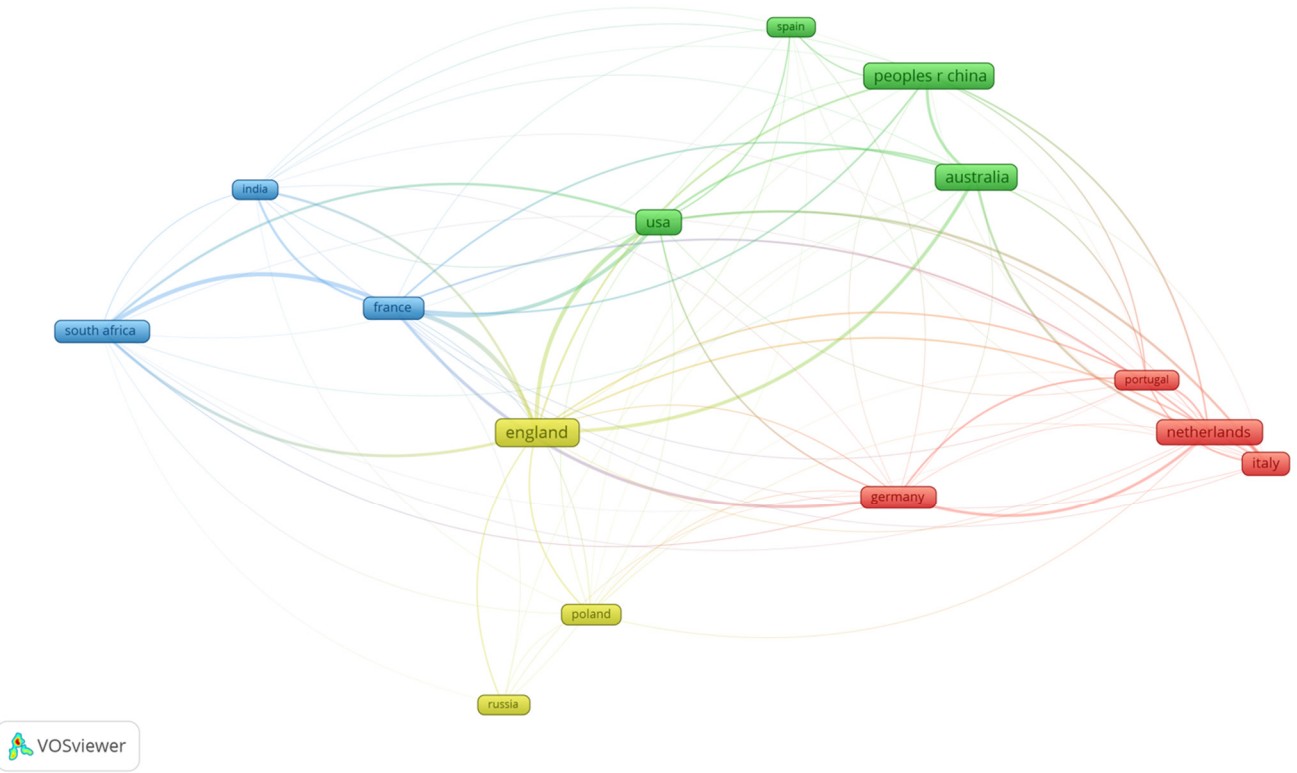

**Figure 3.** Network of countries of affiliation, using bibliometric coupling.

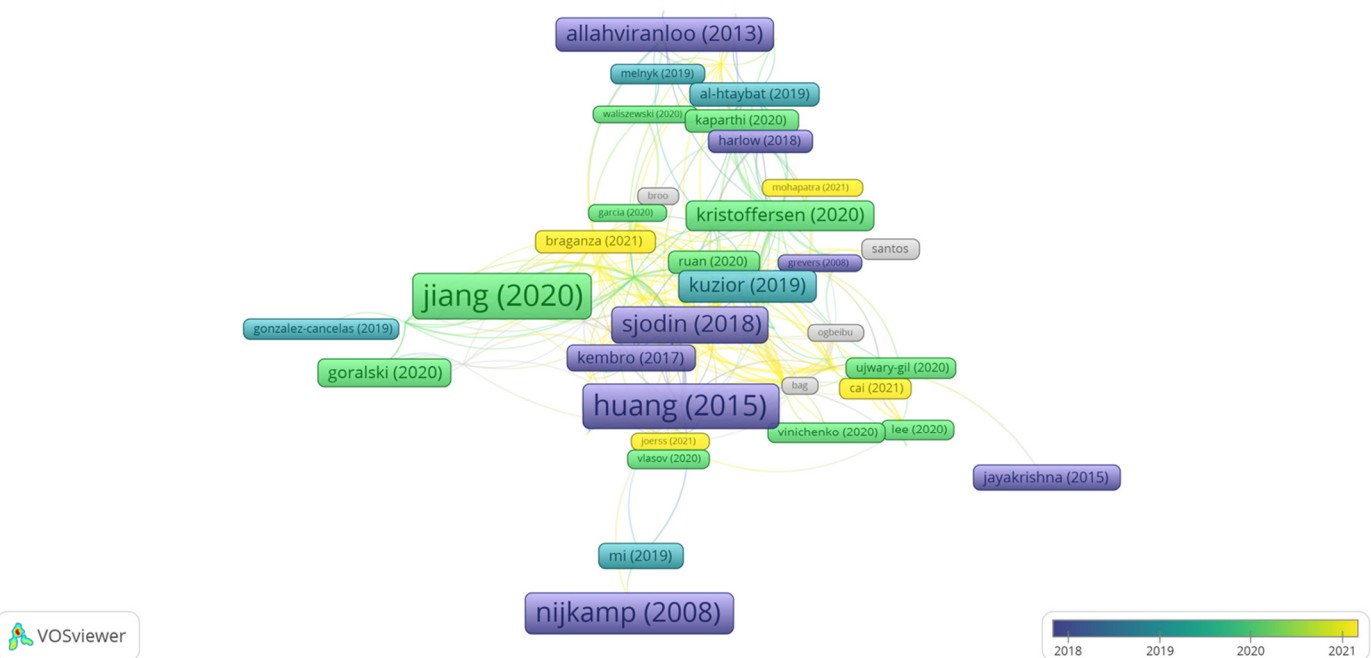

**Figure 4.** Network of related documents using bibliometric coupling. Note: The size of the frame represents the number of similar citations.

### 3.2. Citation and Co-Citation Analysis

The citation analysis allows scholars to understand the relationship among article publications [39] and to identify the most influential publications in a research field. Figure 5 exhibits the most influential articles, where some examples, such as [30,40,41] stand out from the rest. These articles focus on the implementation of AI technologies within companies in the transition towards the Fourth Industrial Revolution.

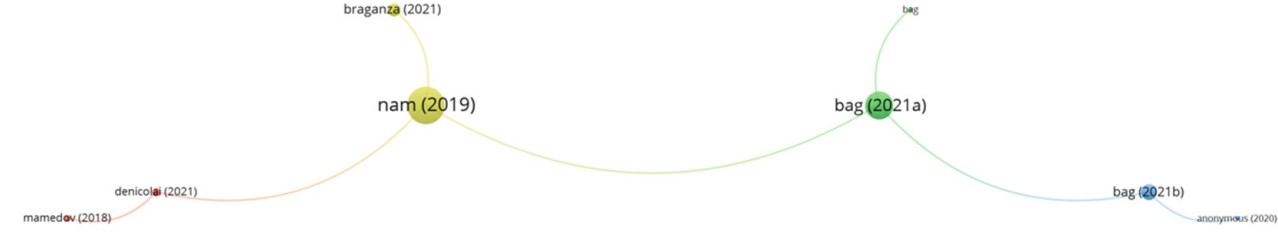

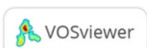

**Figure 5.** Citation network of articles.

Co-citation analysis represents the number of times two secondary articles are cited together in a third article. The Figure 6 examines how often secondary journals are co-cited and reveals the network composition of journals where research is published. In each cluster, it is possible to uncover the leading journal [42]. In the red cluster, Journal of Cleaner Production is the most cited source, which includes articles mainly dedicated to sustainable issues. The green cluster reflects sources from the fields of management and marketing, where Journal of Business Research and Tourism Management are the

most prevalent, in terms of citations and networks established. The blue cluster illustrates strategy and management journals, with Strategic Management Journal emerging as one of the most relevant. A final yellow cluster can also be visualized, with Technological Forecasting and Social Change prevailing in this niche group, intending to link technology to social issues.

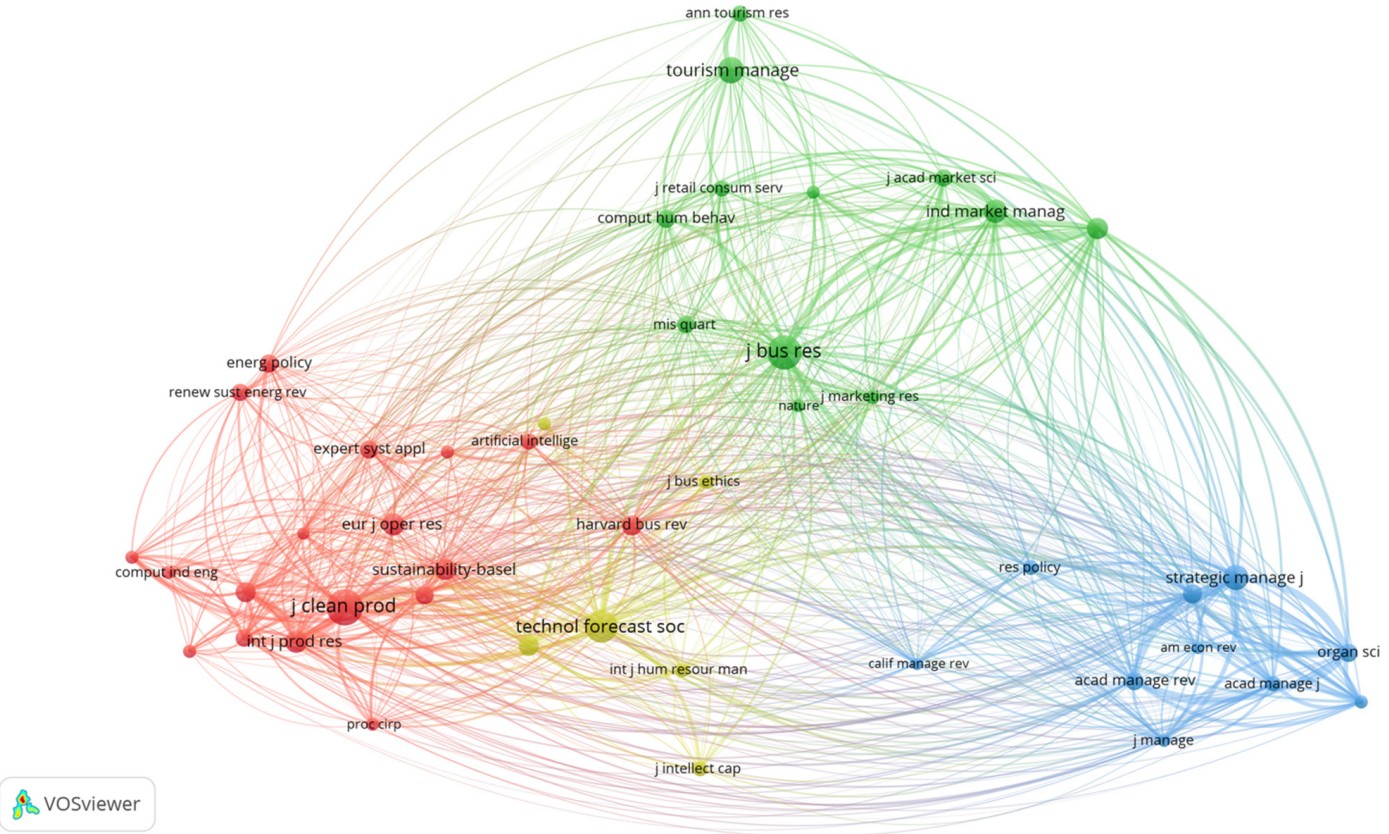

**Figure 6.** Co-citation network of journals. Note: The circles in the figure represent the cited reference and the size of the circle denotes the total link strength of respective cited reference.

### 3.3. Co-Occurrences of Keywords

A keyword co-occurrence network represents the relationships between keywords, which reflect the main context in the literature (see Figure 7). The most prominent node is *artificial intelligence* (AI). This node links with others in the same cluster, such as *internet of things, big data, sustainable development,* or *automation.* The yellow network intersects the blue one, through the *management* and *performance* terms. The green network aggregates words such as *circular economy* and *industry 4.0,* with others associated with AI, e.g., *big data analytics, machine learning* or *systems.* Finally, the red network links the word *sustainability* with *VR* and *AR technologies, innovation,* or issues related to the adoption of such technologies.

Most of the articles analyzed are related to AI (N = 123, 88.5%). The articles on VR and AR (N = 16, 11.5%) links the use of VR and AR technologies to support and promote sustainable concerns in enterprises, tourism, stores, or education [29,43–45]. The adoption of VR, AR and related technologies facilitates distant contacts in real time, consumer decision, and can reduce the costs of travel or new product development [36,43,46]. Retailers and educational institutions can both benefit with the use of VR and AR: the former can use it for framing advertisement messages, to promote sustainable products [45], while educationally, teachers should consider the intersections of design thinking and emerging technologies, for students to engage with the sustainability theme [44]. Benefits for the tourism sector are suggested in studies concerned with the destruction of destination

sites [28,29,47]. The usage of VR/AR technologies can control the overcrowd [48], for instance, when tourists assist a VR immersive visualization, instead of being there, as they have recently provided access to visitors, during the lockdown periods caused by COVID-19 [49].

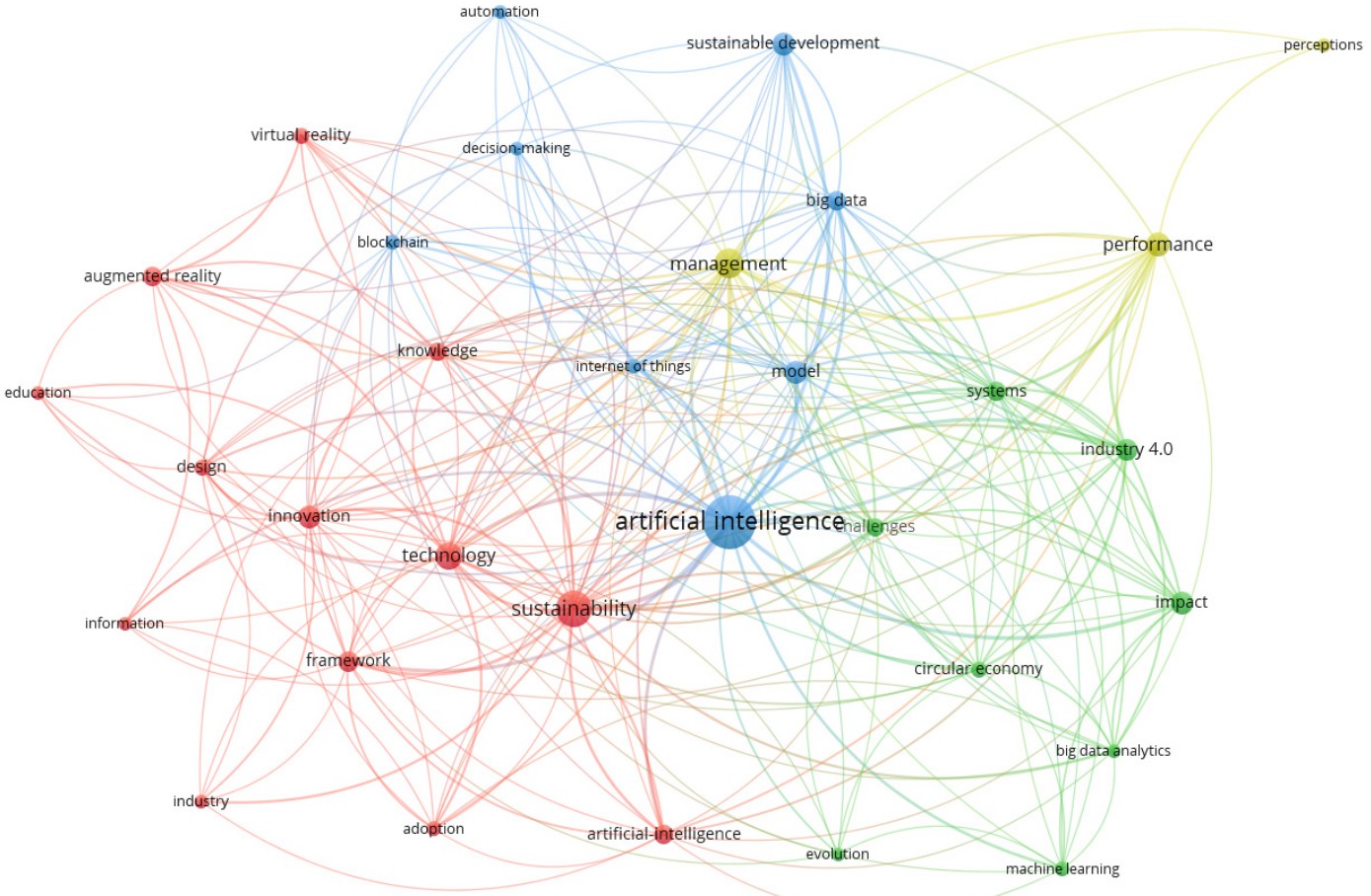

**Figure 7.** Keyword co-occurrence network. Notes: (1) Each node in a network represents a keyword wherein: size of the node indicates the occurrence of the keyword (i.e., the number of times that the keyword occurs); (2) the link between the nodes represents the co-occurrence between keywords (i.e., keywords that co-occur or occur together), (3) the thickness of the link signals the occurrence of co-occurrences between keywords (i.e., the number of times that the keywords co-occur or occur together), (4) the bigger the node, the greater the occurrence of the keyword, and (5) the thicker the link between nodes, the greater the occurrence of the co-occurrences between keywords. Each colour represents a thematic cluster, wherein the nodes and links in that cluster can be used to explain the theme's (cluster's) coverage of topics (nodes) and the relationships (links) between the topics (nodes) manifesting under that theme (cluster).

Although the first articles aggregating AI with sustainability, in the analyzed database, were published back in 2008 and 2013 [50], it is since 2017 that we assist to an explosion of publications about the Fourth Industrial Revolution. These are mainly focused on industry-level applications but represent the usage of technology to create an inclusive, human-centered future culture [51]. Thus, leaders, employees and citizens need to create a new organizational culture to incorporate AI technology, taking into consideration the sustainability issues and the well-being of society [51,52].

The incorporation of AI in industry and society creates benefits because it facilitates better resources allocation, treating large amounts of data, which allows a focus on what customer really desires, and to reduce waste. With AI-enabled agents (robots or otherwise) performing traditional jobs and tasks, a reformulation of labor and of the whole society, can be expected. In 2019, some publications started to discuss the ethical concerns and

risks, resulting from the incorporation of AI algorithms and agents in our industry and society [53]. Furthermore, the year of 2020 is marked by the discussion of the AI impact on the United Nations (UN) Sustainable Development Goals (SDGs). The fact that some countries are more open culturally and economically than others to the implementation of AI systems, creates new inequalities [54,55] widely discussed in literature.

## 4. Discussion

*4.1. Main Research Fronts on Technology, Sustainable Development and Tourism*

For examining the field's intellectual structure, informed by the most relevant and emerging topics of discussion among scholars, latest contributions were reviewed, found in the following subject areas from the dataset: (i) tourism sustainable development; (ii) development of breakthrough technologies which may impact tourism; and (iii) theoretical frameworks or enabling factors, which can facilitate their implementation. Based on a systematic review of these articles, we have uncovered five topical groups, which we suggest that will become the major work streams in the future. Our analysis shows that only a small portion of the most recent publications do, in fact, directly address the tourism sector. However, we consider that a wider number of studies, originated from other business or scientific areas, provide useful contributions and will influence the direction of research towards a broader perspective on how breakthrough technologies may shape the future of sustainable tourism. Effectively, as shown on Table 1, only one from these work streams addresses tourism-related phenomena (accounting for only 9% of publications from 2021), while the others examine either supporting theories/frameworks, influencing drivers and barriers, or the outcomes of technology adoption.

**Table 1.** Main research streams.

| Main topics | Theories and Frameworks | Stakeholders | Industries | References |
|---|---|---|---|---|
| (1) Review of theories, frameworks, and literature on new technologies' role and accelerating the innovation roadmap | | | | |
| STARA (Smart Technologies, AI, Robotics and Algorithms), environmental dynamism, GOIE (Green Organisational Innovative Evidence), crowdsourcing, open innovation, Big Data | TCT (Transaction Cost Theory), Stakeholder theory, Triple Helix model, Knowledge-based view, IAS (Innovation-Automation-Strategy) cycle | Organisations and Companies, Society (Government-University-Industry) | Transversal | [33,34,56–59] |
| AI, co-creation, data mining, unsupervised machine learning algorithms | DWA (Design Weltanschauung Model), grounded-theory | Companies, Consumers | Food | [60,61] |
| AI/XAI (explainable AI), Machine learning, FinTech | Neural networks, FRST (Fuzzy-Rough Set theory), MCDM (Multiple Criteria Decision Making) | General Society, Organisations and Companies | Finance and Banking | [62–65] |
| (2) The impact of new technology adoption for Sustainable Development and Social (In)equality | | | | |
| AI (AI readiness, Trustworthy AI), Big Data, IoT, SD, SDM (Single Digital Market), egalitarian societies, industrial robotics, job trust, employee engagement, Sharing Economy | Labour theory of value | Countries and General Society, Employees | Transversal | [41,66–71] |
| AI, carbon footprint | | Farmers | Agriculture | [72] |
| AI, Machine learning, Big Data | | Organisations and Companies | Finance and Banking | [73] |

**Table 1.** *Cont.*

| Main topics | Theories and Frameworks | Stakeholders | Industries | References |
|---|---|---|---|---|
| (3) The benefits of digital transformation for strategy, competitive advantages, and reputation management | | | | |
| AI, Social listening, Social and online media, reputation management, anti-trust, price discrimination, big data/market intelligence, internationalization, digitalization, sustainability-readiness | Readiness lifecycle models | Organisations and Companies | Transversal | [74–76] |
| AI, Social and online media, content curation, machine learning | | Companies and Prosumers | Digital media and advertising | [32] |
| (4) Adoption of new technologies: barriers, drivers, and perceived added value | | | | |
| Digital humans, AR, VR | | Consumers | Fashion | [76,77] |
| AI, technology adoption | | Farmers | Agriculture | [78] |
| AI, big data, circular economy, institutional pressures | Resource-based theory, Institutional theory | Organisations and Companies | Automotive | [40] |
| (5) Technology as enabler of H&T sustainable development, competitiveness and customer experience | | | | |
| AI, ANN (Artificial Neural Networks), designing collaborative strategies, co-creation, service agents | Auto-contractive map, Behavioural Reasoning theory, fsQCA (Fuzzy-set qualitative comparative analysis) | Companies and Entrepreneurs, Consumers | Tourism | [10–13,38,79] |

After summarizing relevant issues, contributions, and supporting theories, some of the promising trends and intriguing questions which emerge from literature are highlighted next.

4.1.1. Review of Theories, Frameworks, and Literature on the Role of New Technologies and Accelerating the Innovation Roadmap

In this research stream, authors mainly develop frameworks for a projective view on the most suitable capabilities, preconditions, and practices, required for achieving a future innovation roadmap, for entrepreneurial agents [56], crowdsourcing innovators [58], and green organisational initiatives [34], as well as theorizing on the relations between process, technology, and management innovations [59]. In specific, the future opportunities for finance, auditing, and banking systems are widely discussed among scholars. Some of the bigger challenges identified [64,65]—which may also impact tourism—relate to the shift from inventions towards applications, with AI becoming a business requirement instead of an optional strategy. More efficient processes will benefit consumers and create new occupations, while posing formidable challenges to the maintenance of existing jobs. Another stimulating debate also emerges between 'black' and 'white box' approaches [62]: while the former group values deep learning and computational efficiency over human interpretability, the latter employs traceable methods to enhance AI explainability to human decision-makers.

Furthermore, based on the most recent contributions from scholars more dedicated to the benefits of technology for overall business sustainability [33,57], we can expect future advancements on demonstrating transversal organisational gains, such as cost reduction and augmented decision-making capabilities provided by big data analytics. The potential benefits arising from co-creation opportunities, AI and machine learning are suggested [60,61], which may disrupt the competitive landscape for food industry and hospitality, per example, by enabling players with proactive flexibility for more robust marketing applications and supply chain competitive advantages, because of demand forecasts driven by big data and more interactive relationships with customers.

### 4.1.2. The Impact of New Technology Adoption for Sustainable Development and Social (In)equality

We highlight three inter-connected levels of discussion, which will play a crucial role in shaping the future discussion on how technology may (or may not) contribute to a more sustainable and fairer world: (i) the challenges for labour—with evidence showing that economic gains from using industrial robots are realized only in more developed nations—while, at the same time, new job trust and employee alienation issues arise [41,66]; (ii) the impacts of Industry 4.0 for value creation and 'smart services' implementation, altering the producers–workers–consumers dynamics, but with rising concerns about the risks of aggravated global inequalities, in the era of AI and automation [69,71]; and, finally, (iii) the role of digital transformation for achieving SDGs, related to agriculture's carbon footprint, productivity gains, and inclusion in new digital economic ecosystems [67,68,72].

### 4.1.3. The Benefits of Digital Transformation for Market Competition, SMEs' Strategy, and Reputation Management

The outcomes of AI-driven disruption for online media and digital marketing purposes, will increasingly impact the tourism sector, as well as through the ability of firms to, for example, gather more information about consumer behaviours via social listening, to create predictive models and to manage company reputation [32,76]. The greater analytical sophistication of AI can also modify the future market structures, by influencing companies' conduct and patterns of competition [75]. Three growth paths are suggested for SMEs: internationalization, digitalization, and sustainability [74], with the latter highlighted as the key future competitive driver for firms with international presence, which will facilitate the realization of the other two paths.

### 4.1.4. Adoption of New Technologies: Barriers, Drivers, and Perceived Added Value

In this research stream, suggestions are given for a future business roadmap to augment the channels of interaction to enhance their levels of service and customer experience, and stay competitive in a post-COVID, constantly evolving marketplace. Customer willingness to be engaged through such virtually 'humanized' touchpoints, is shown to depend on demographic factors, key traits of 'digital humans', form and personal device of interaction [77], based on empirical evidence obtained in the fashion category.

Conversely, on the prosumer side, farmers and wine producers were examined as examples where the quality level of products depends more on traditional norms than on disruptive innovations. In this case, the lack of knowledge and strategic focus constitute the main barriers, towards accelerating the digital business transformation, as is suggested across other industries [78]. For factories, such as in the automotive sector, digital transformation can enable the development of circular economy capabilities, thus contributing to more sustainable manufacturing practices; however, companies will need to invest in workforce skills combined with tangible resources [40]. Besides reinforcing workforce skills and digital literacy, the understanding of the main motives for consumers to engage with 'digital humans' is a critical success factor, which will emerge for tourism operators in the nearby future.

### 4.1.5. Technology as Enabler of H&T Sustainable Development, Competitiveness, and Customer Experience

The current study demonstrates the lack of research addressing the specific challenges and impacts of technology adoption and environmental sustainability for tourism. A notable exception [79] reports on AI's important role in environmental modelling, such as agriculture, water quality, landscape, and forest modelling, which can shape the way environmental impacts are forecasted and investigated in the context of sustainable tourism development. Some other recent studies elaborate on the role of technology for supporting more sustainable business models, for touristic destinations and stakeholders, which could otherwise be a saturated market, severely affected by the effects of COVID-19.

Creating more engaging, competitive tourist experiences (such as derived from co-creation platforms) is one of possible future directions, as long as individual entrepreneurs are willing to cooperate and adopt technological innovations [12]. The perceived value of co-creation is also promising on the consumer side and is confirmed as antecedent of tourist behavioural intention. Interesting findings here suggest that, while consumer barriers are commonly related to their unwillingness to give up old habits, drivers are context-dependent [11] with functional value and added convenience, per example, presenting opportunities for AI-enabled personal assistants to attract customers looking to save time on routine tasks (e.g., searching for flights and accommodation).

Yet, other stakeholders will need to be involved in the future, bringing on board actors from educational institutions, adjacent or even non-related industries, for the fulfilment of public policies, at urban/regional or national levels. This broader collaboration will integrate destination competitive strategies into a wider economic policy, which is essential to overcome ad hoc opportunistic behaviours from individual agents [10], another bottleneck which often compromises the realization of sustainable development opportunities for tourism.

To support the post-pandemic recovery in H&T, research should focus on determining to what extent are industry recovery programs aligned with the evolution of travellers' expectations. Some new market demand trends are projected, such as traveller wellness, contactless services, and pro-environmental practices [38], therefore forthcoming studies can benefit from deploying big data and data mining techniques, to issues related to AI and robotics, eco-tourism, hygiene, health care and wellness. Evidence is available showing that consumers' intention to use social robots in H&T services, such as restaurants [13], may depend on perceived value and human-robot interaction dimension, namely, empathy and information sharing. But at what level will this change of paradigm provide a better opportunity for higher-value tasks to employees, per example, is one of the many intriguing questions still to be determined.

### 4.2. Future Research Agenda

In terms of future avenues of research, a set of open questions is offered, which emerge from examining insights and integrating reflections spread across multiple subject areas, explored throughout the literature review. Our firm suggestion for managers, leaders and scholars, is to observe some possible impediments, and consider them when directing their own efforts, in a way that can contribute to solving–instead of aggravating–them: (a) lack of inter-disciplinary academic collaboration; (b) lack of cross-industry knowledge sharing; (c) lack of focus on SMEs; (d) lack of focus on the more digitally-excluded demographic and ethnical groups (e.g., applicable to customers, workers, local H&T entrepreneurs, etc.); (e) lack of empirical applications in under-developed nations; (f) lack of available data and dedicated theories, considering the recency of technological applications in the H&T context.

Transversal research questions:

1. How to create trust in the adoption of new technologies, breaking old habits and building a better comprehension of its merits?
2. What are the reasons for/against adopting co-creation platforms, and how will these platforms gain relevance in the practices of H&T stakeholders (e.g., *consumers, entrepreneurs, service agents, workers)*?
3. How will the creation of global digital ecosystems (such as the new EU digital market) relate to the achievement of sustainable development goals in H&T?

Questions about customer engagement:

4. What is the relationship between travellers' values, beliefs, profiles, and their behavioural intentions towards using AI-enabled travel service agents (e.g., *chatbots)*?
5. What are their attitudes towards, and/or propensity to interact, with digital humans? What are their preferred ways of interaction?

6. What are expectations and/or propensity to interact with VR and AR-enabled experiences?

Questions at nationwide, regional or destination levels:

7. How will technology boost destinations' innovativeness and competitiveness, leveraging enriched customer experiences? Which areas of impact and digital innovations should be prioritized?
8. Which stakeholders should be involved in the process of designing collaborative strategies?
9. Which factors can enable/inhibit AI adoption? How do they vary across regions or type of destination?

Questions at the organizational level:

10. How to make AI adoption compatible with promoting a more productive workplace, a better employer engagement, and more decent labor practices in H&T?
11. What role can big data play for enhancing the marketing performance of H&T operators, or other related service providers?
12. What is the relationship between digitalization and internationalization of tourism agents, and the effects of their sustainability-readiness?
13. What new AI applications in the social media and digital advertising ecosystems, including automated, algorithm-based online conversation analysis, can support the post-pandemic turnaround and reputation management of H&T firms?
14. How can leader STARA competences, green creativity components and environmental dynamism aid tourism organizations to boost their GOIE?
15. What are the critical success factors for entrepreneurs/SMEs to be aware and trust on AI benefits, without compromising their identity?

*4.3. Theoretical Implications*

The purpose of this study was to examine the main contributions and latent trends in the pertinent literature related to H&T, technology, and sustainable development, providing a clear picture of the future possibilities of the Fourth Industrial Revolution for sustainable tourism. For that effect, we offer a bibliometric analysis, followed by a comprehensive review of the most influential contributions, for producing meaningful insights and connect the areas of sustainable tourism and technological innovation.

Results show that these domains are not frequently related in literature. A sample of 139 peer-reviewed articles is examined, and the main study topics observed include: (i) how organisations should leverage the adoption of technological innovations, by setting new strategies, preparing core processes, structures, job definitions, and developing new capabilities; (ii) concerns about ethical, labour and social equality issues, regarding the shared role with 'AI-driven robots' and 'digital humans' in society; (iii) H&T post-COVID economic sustainability and future competitive challenges.

The most influential contributions and contributors are highlighted, and the collaborative network is analysed. Four journal clusters are identified in our co-citation analysis, namely focused on sustainability, clean energies, and industries; management and marketing; corporate strategy; and the connection of technology to social issues. Overall, AI is the main technological trend observed (e.g., in 88.5% of papers), growing exponentially since 2017, particularly through industry-level applications and its impacts on a future, more inclusive global culture. The merits of AR/VR for sustainable development, business and education are also explored, including the mitigation of over-tourism and protection of destinations' natural ecosystems, by providing alternative experiences to tourists. Subsequently, the intellectual structure of literature is categorised into five research streams, identified by the present study, which inspired our proposed toolkit of future considerations and research questions.

This study contributes in a number of ways to the theoretical development of sustainable tourism and related fields. To the best of our knowledge, this is one of the earliest studies to employ bibliometric and systematic review techniques, to examine the future

impacts of technology on a specific industry. Moreover, it crosses over inter-disciplinary boundaries to identify the most influential and promising work streams, summarizing findings and contradictions. Finally, a future agenda is proposed, formed by a set of open questions and possible impediments, stemming from the most recent trends uncovered in this study, which may help future scholars to guide their own research efforts, and further advance collective knowledge in this area.

### 4.4. Managerial and Social Implications

Our findings have significant implications for national authorities, public or private organisations promoting pro-environmental tourism or striving for the business sustainability of the sector altogether. The proposed future agenda can inspire the creation of a transformation roadmap for sustainable tourism, suitable for stakeholders navigating the challenges ahead. The breadth of different frameworks and models used—regarding the implementation and outcomes of disruptive technologies—can appear overwhelming, but we offer a summary (Table 1), which highlights the main contributions, categorised by topical orientation (*research fronts*) and stakeholder type. H&T entrepreneurs or decision-makers, sustainability experts and technology consultants can find here referenced works, which may be used to expand on their insights for informing their own diagnose, reflecting on the level of digital readiness and environmental dynamism, and defining a strategic plan for managing a successful transition on their organisations of influence. Destination site, hotel or other tourism-related managers can further explore the critical success factors examined to design new competitive advantages. However, and although the digital revolution can provide new, exciting opportunities, before managers set in motion any strategic change or innovation initiative, they should carefully consider building trust first, by reinforcing the level of digital literacy of workers and other key stakeholders, as well as identifying and securing access to the required resources. From a practical standpoint, future interventions should cultivate buy-in throughout the introduction phase of any new tool, able to substantially affect the daily tasks of H&T employees, by instilling a sense of co-ownership, and by developing tailor-made training programs addressing their actual work context or location sites.

With regard to the use of VR/AR/AI technologies to appeal to tourists or prospect customers, results from our review call for marketeers to define first what are the tangible benefits for end-users (e.g., interactivity and sensorial stimulation; convenience, time saving or availability), and how to express them in their communication messages to resonate with their audience and persuade them to engage through the intended virtually enhanced touchpoints. An extensive testing phase is recommendable–using both internal and external volunteers–with needs and openness towards digital interactions similar to the target audience. This should be done at a point where relevant adjustments or improvements can still be made—during the co-creation phase—and involving all critical technical solution providers.

### 4.5. Limitations of the Systematic and Bibliometric Process

Despite its contributions, the present study is not without limitations. First, the analysis used is based on bibliographic metadata, mainly number of publications, citations or co-citations, and frequency of keywords, which means that some topics might have received comparatively more attention than others, disregarding their actual significance. Second, as in any literature review, the search query could be improved: papers which did not mention the selected terms explicitly in their title, abstract or keywords, may have been excluded. We encourage future scholars to broaden the search criteria, by including similar or related terms (e.g., *pro-environmental tourism*), terms expressing other technological innovations, or any new term which can emerge in the future. Third, as our dataset covers only peer-reviewed articles, other kind of published material, such as conference proceedings, may be used to identify additional topics relevant to our present theme. Moreover, even though no publication timeline restriction was imposed,

papers explored for the purpose of discussing future trends and research questions, were mainly from the 2020–21 period. Some previous, but relevant documents, may have been unintentionally ignored during this process. Lastly, additional bibliometric network analysis enrichment techniques could have been employed, such centrality metrics or strategic diagrams. Although these limitations can be considered in future efforts, we hope that this study can help as a beacon for researchers and practitioners, to direct their works in a more focused and productive way. Empirical case-studies, for example, could be used for in-depth observations of the technology implementation process in the H&T context, and its outcomes for sustainable development.

## 5. Conclusions

Recently, environmental sustainability has become a topic of interest due to the accelerated rate of environmental degradation and climate change, capturing the public attention, and fostering intense debate among the scientific community, media, and political stakeholders. On a parallel route, disruptive change driven by technological innovations (such as AI, Big Data, IoT, VR and AR), is now a topic of growing importance for researchers and practitioners, considering its many applications in diverse domains, and leading to conflicting views on the merits and risks from technology adoption, regarding social inclusion and labour imbalance, per example.

Although the H&T industry is considered the largest worldwide [1,2] and accountable for one of the heaviest carbon footprints [5], yet continues to exhibit a considerable knowledge gap concerning drivers and barriers of pro-environmental behaviours [80,81]. Moreover, contrary to expectations and despite the potential benefits, the impacts of technology on sustainable tourism development appear to be so far neglected in the network of academic publications. Findings from this study's bibliometric analysis and systematic review suggest that the outcomes of AR, VR and AI adoption, in the context of tourism's sustainable development, are only scarcely examined. Five research fronts are revealed and described, which represent the state-of-the-art in literature addressing technology's benefits and risks for nations' sustainable development, companies' business competitiveness, as well as the main factors influencing the adoption by consumers, prosumers, organisations, and enabling frameworks used for observing implementation.

In summary, tourism faces great challenges for economic turnaround, and for unleashing the full potential of technological innovations. When it succeeds in making a swift transition, leveraging these new added value opportunities, and evolving into more environmentally and socially sustainable business models, the sector will be at the forefront of new service capabilities and solutions. This encloses the opportunity for academic researchers to contribute to theory development and to foster positive changes in managerial practices in areas such as hotel marketing and management.

**Author Contributions:** Conceptualization, S.M.C.L. and J.N.; methodology, S.M.C.L.; software, S.M.C.L.; validation, J.N. and S.M.C.L.; formal analysis, S.M.C.L.; investigation, J.N.; resources, J.N..; data curation, J.N.; writing—original draft preparation, J.N.; writing—review and editing, J.N.; visualization, J.N.; supervision, S.M.C.L.; project administration, S.M.C.L.; funding acquisition, J.N. All authors have read and agreed to the published version of the manuscript.

**Funding:** This research was funded by ISCTE Business School's Merit Scholarship for PhD Students.

**Institutional Review Board Statement:** Not applicable.

**Informed Consent Statement:** Not applicable.

**Data Availability Statement:** Not applicable.

**Conflicts of Interest:** The authors declare no conflict of interest in the present study.

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
