# Peer review of "Shaping a View on the Influence of Technologies on Sustainable Tourism"

_sustainability, doi:10.3390/su132212691_

Round 1

Reviewer 1 Report

Dear authors,

I find your paper very interesting and relevant for the tourism area, and very important in the current context, with a high orientation towards sustainability.

The paper is well structured and properly developed. However, I have a series of minor suggestions that can add value to your paper:

  • the methodology section should be improved and expanded, outlining more information regarding the data retrieval process and selection criteria.
  •  the conclusions section could be more detailed, especially by adding some key information from the results and discussion sections.

Author Response

Dear Reviewer,

Thank you for giving us the opportunity to submit a revised draft of the manuscript “Shaping a view on the influence of technologies on Sustainable Tourism” for publication in Sustainability's forthcoming special edition about Technological Transformations towards a More Sustainable Environment in Hospitality and Tourism.

We are grateful for the insightful comments and valuable improvements suggested to our manuscript and appreciate the time and effort that you dedicated to providing feedback. We have incorporated the suggestions made. Those changes are highlighted within the manuscript, using 'track changes'. Please check below, in blue, our point-by-point response to your concerns and comments.

  1. the methodology section should be improved and expanded, outlining more information regarding the data retrieval process and selection criteria.

[Authors' response]: Thank you for pointing that out. We agree with the reviewer. Materials and Methods' section was expanded and organized into two separate sub-topics: search protocol/data collection and data analysis. Further details are now provided for each sub-topic, so that readers can appreciate (and replicate, if necessary) the procedures and techniques used, including a diagram displaying the search steps in detail.

  1. the conclusions section could be more detailed, especially by adding some key information from the results and discussion sections.

[Authors' response]: Although the indication in the journal guideline template mentions this section as optional - "This section is not mandatory but can be added to the manuscript if the discussion is unusually long or complex" - the 'Conclusions' section is now expanded, as suggested by both reviewers, summarizing the key findings. Reflection upon the study's aims and how they were achieved, major outcomes and limitations, as well as their both implications, are still included in 'Discussion'.

Reviewer 2 Report

This study presents an overview of the literature regarding sustainable tourism and technological innovations. The topic is appealing and offers a new perspective on the possibilities of tourism development based on the application of the new technologies. I suggest to the authors to perform some improvements.

  1. I propose to the authors to merge the "Methods and materials" and "Results" in one section because the “Methods and materials” are very briefly explained.
  2. The authors should better emphasize the research intention in the "Introduction section". Please, indicate what you did and why you did it.
  3. The figures in the paper are not clear and there are very small. It will be good if there is a possibility to improve their quality.
  4. Please, extend the conclusion. Clearly emphasize the key findings.

Author Response

  1. I propose to the authors to merge the "Methods and materials" and "Results" in one section because the “Methods and materials” are very briefly explained.

[Authors' response]: Thank you for pointing that out. Following the other reviewer's suggestion, Materials and Methods' section was expanded. It is now organized into two separate sub-topics: search protocol/data collection and data analysis, with a detailed diagram displaying the search procedures step-by-step.

  1. The authors should better emphasize the research intention in the "Introduction section". Please, indicate what you did and why you did it.

[Authors' response]: We fully agree and have now improved the 'Introduction' section' in line you with your comment.

  1. The figures in the paper are not clear and there are very small. It will be good if there is a possibility to improve their quality.

[Authors' response]: The images were inserted again, with maximum picture resolution, and larger size. The original files will be provided if necessary.

  1. Please, extend the conclusion. Clearly emphasize the key findings.

[Authors' response]: Although the indication in the journal guideline template mentions this section as optional - "This section is not mandatory but can be added to the manuscript if the discussion is unusually long or complex" - the 'Conclusions' section is now expanded, as suggested by both reviewers, summarizing the key findings. Reflection upon the study's aims and how they were achieved, major outcomes and limitations, as well as their both implications, are still included in 'Discussion'.

Round 2

Reviewer 2 Report

The authors successfully met all the requirements. Well-done.